# Engineering an Optimal Y280-Lineage H9N2 Vaccine Strain by Tuning PB2 Activity

**DOI:** 10.3390/ijms24108840

**Published:** 2023-05-16

**Authors:** Se-Hee An, Seung-Min Hong, Jin-Ha Song, Seung-Eun Son, Chung-Young Lee, Kang-Seuk Choi, Hyuk-Joon Kwon

**Affiliations:** 1Laboratory of Avian Diseases, College of Veterinary Medicine and BK21 PLUS for Veterinary Science, Seoul National University, Seoul 08826, Republic of Korea; 2Research Institute for Veterinary Science, College of Veterinary Medicine, Seoul National University, Seoul 08826, Republic of Korea; 3Department of Microbiology, School of Medicine, Kyungpook National University, Daegu 41566, Republic of Korea; 4Laboratory of Poultry Medicine, Department of Farm Animal Medicine, College of Veterinary Medicine and BK21 PLUS for Veterinary Science, Seoul National University, Seoul 88026, Republic of Korea; 5Farm Animal Clinical Training and Research Center (FACTRC), GBST, Seoul National University, Pyeongchang 25354, Republic of Korea; 6GeNiner Ltd., Seoul 08826, Republic of Korea

**Keywords:** avian influenza A virus, H9N2, recombinant vaccine strain, mammalian pathogenicity, bivalent oil emulsion vaccine

## Abstract

H9N2 avian influenza A viruses (AIVs) cause economic losses in the poultry industry and provide internal genomic segments for the evolution of H5N1 and H7N9 AIVs into more detrimental strains for poultry and humans. In addition to the endemic Y439/Korea-lineage H9N2 viruses, the Y280-lineage spread to Korea since 2020. Conventional recombinant H9N2 vaccine strains, which bear mammalian pathogenic internal genomes of the PR8 strain, are pathogenic in BALB/c mice. To reduce the mammalian pathogenicity of the vaccine strains, the PR8 PB2 was replaced with the non-pathogenic and highly productive PB2 of the H9N2 vaccine strain 01310CE20. However, the 01310CE20 PB2 did not coordinate well with the hemagglutinin (HA) and neuraminidase (NA) of the Korean Y280-lineage strain, resulting in a 10-fold lower virus titer compared to the PR8 PB2. To increase the virus titer, the 01310CE20 PB2 was mutated (I66M-I109V-I133V) to enhance the polymerase trimer integrity with PB1 and PA, which restored the decreased virus titer without causing mouse pathogenicity. The reverse mutation (L226Q) of HA, which was believed to decrease mammalian pathogenicity by reducing mammalian receptor affinity, was verified to increase mouse pathogenicity and change antigenicity. The monovalent Y280-lineage oil emulsion vaccine produced high antibody titers for homologous antigens but undetectable titers for heterologous (Y439/Korea-lineage) antigens. However, this defect was corrected by the bivalent vaccine. Therefore, the balance of polymerase and HA/NA activities can be achieved by fine-tuning PB2 activity, and a bivalent vaccine may be more effective in controlling concurrent H9N2 viruses with different antigenicities.

## 1. Introduction

H9N2 avian influenza viruses (AIVs) pose a threat to the poultry industry, leading to decreased egg production and increased mortality when complicated with other pathogens. Since the first outbreak in 1994, these viruses have diverged into several lineages, including Y439/Korea in Korea, Y280/G9 in China, and G1 in Egypt [1,2,3,4,5]. The Y439/Korea-lineage was first reported in 1996 and was initially eradicated, but it became endemic again after its recurrence in 1999. In 2007, a large-scale inoculation of an oil emulsion vaccine composed of the early Y439/Korea-lineage virus [A/Chicken/Korea/01310E20/2001(H9N2), 01310CE20] was carried out, leading to few reports of H9N2 in vaccinated layer farms [4,5,6]. However, several reassortant H9N2 genotypes have been isolated from unvaccinated Korean native chickens, ducks, and other poultry in live poultry markets (LPM). Since the first reports in LPM, 2020, Y280-lineage H9N2 viruses, which are more pathogenic than Y439/Korea-lineage viruses, have become prevalent in poultry farms and have been traced back to a most likely progenitor (A/chicken/Shandong/1844/2019) in China [7,8,9,10].

Wild aquatic birds serve as natural reservoirs for AIVs, and the transmission between different flocks and species may occur mainly through contaminated water that is shared by various wild birds. Chickens, which are highly susceptible and raised in close, densely populated spaces, play a role in amplifying the virus population and hastening the selection of viral quasi-species with better fitness. The Y280 and G1-lineages of H9N2 viruses have evolved into multiple sub-branches and clusters with high nucleotide and amino acid variability, and they are still being isolated from poultry [11]. As a result, chicken-adapted H9N2 viruses play a key role in providing competent internal genomic segments for newly introduced wild bird AIVs, allowing for rapid adaptation and increased pathogenicity to chickens and humans, such as the H5N1 and H7N9 AIVs [12,13,14].

To date, various mammalian pathogenicity-related mutations (MPMs) have been reported in different genes of AIVs. However, only a few mutations have been proven to have been acquired during mammalian infections, such as E158G, G591R/K, E627K, and D701N in the PB2 gene [12,15,16]. The E627V mutation in the G1-lineage virus from Egypt is suspected to have been acquired in an unknown mammalian host, but, for the majority of the MPMs identified in H9N2 viruses, it is unknown whether they were acquired in mammalian hosts or not [1,17,18,19,20]. The Y280 and G1-lineage viruses share a Q226L mutation in hemagglutinin (HA) that increases the affinity for the mammalian receptor (α2,6-sialogalactose) and cumulative mutations in the 370- and 400-loops of the secondary sialic acid-binding site (2SBS) that reduce the affinity for the avian receptor (α2,3-sialogalactose) of the neuraminidase (NA) [21,22,23]. These mutations are likely to have been acquired during chicken adaptation and may be the result of balancing HA and NA activity for improved viral fitness [5]. Additionally, minimal essential mutations found in avian hosts have been reported to increase the viral fitness required for replication in mammalian hosts [15,24,25,26,27]. Some of these mutations, such as I66M, I109V, and I133V of PB2 (MVV), are shared by most AIVs and likely increase the integrity of PB2 with PB1 and PA, resulting in increased polymerase activity [15]. To date, the rank order of mutations in PB2 affecting polymerase activity has been reported, which may be useful for the modulation of polymerase activity [28]. Viral fitness is a multi-genic trait, and the balance of polymerase and HA/NA activities may play a crucial role. Therefore, it may be important to determine if the balance can be achieved through PB2 tuning [29,30]. The Y280 and G1-lineage H9N2 viruses have been reported to have mammalian pathogenicity, although fatal infections in humans have never been reported [12,31]. As a result, it is important to prevent H9N2 infections in poultry farms, and various vaccine strains have been developed to achieve this goal [4,32,33,34,35].

A/Puerto Rico/8/1934 (H1N1) (PR8)-derived recombinant vaccine strains for poultry have been developed, containing the HA and NA genes of contemporary field viruses and the six internal genomic segments of PR8. While these vaccine strains showed high replication efficiency in embryonated chicken eggs (ECEs), some grew poorly in the ECEs and showed high mammalian pathogenicity [29,30]. One issue with PR8-derived recombinant vaccine strains is that they contain the PB2 of PR8, which includes multiple MPMs, including the highly potent E627K mutation, which increases the mammalian pathogenicity of the vaccine strain and poses a biosafety risk during vaccine production. To address this problem, the PR8 PB2 was replaced with a non-pathogenic PB2 (01310 PB2) to successfully develop safe and highly productive H5Nx and Egyptian G1-lineage vaccine strains [29,30,36]. In this study, we aimed to generate the optimal vaccine strain for the prevention of Korean Y280-lineage viruses, considering factors such as replication efficiency in ECEs, antigenicity, and mammalian non-pathogenicity. We characterized a virus isolate from 2020 in terms of its genetic makeup and changes in HA, NA, and M2e epitopes. We replaced the PR8 PB2 with the 01310 PB2, which showed decreased virus titer but was recovered by introducing MVV into it. However, the introduction of the L226Q mutation in the HA changed the antigenicity and increased the mouse pathogenicity of the recombinant virus. Finally, we evaluated the effectiveness of a bivalent vaccine compared to monovalent vaccines of the Y439/Korea and Korean Y280-lineage recombinant virus.

## 2. Results

### 2.1. A Korean Y280-Lineage H9N2 Virus (SL20wt) Is Genetically Closely Related to Genotype S H9N2 Viruses in China and Different from Y439/Korea-Lineage Vaccine Strain

SL20wt was found to have the highest nucleotide identity with the H9N2 strains of A/Korean native chicken/South Korea/N20-99/2020 (N20-99) and A/chicken/Shandong/1844/2019 (SD1844) among the available Korean and Chinese H9N2 strains (Appendix A). The nucleotide and amino acid identities to N20-99 and SD1844 were 99.27–99.87% and 98.75–100%, and 98.73–99.74% and 96.70–100%, respectively. SL20wt is classified into the genotype S H9N2 viruses in China along with N20-99 and SD1844 [33,37]. It shares the common MPMs found in most genotype S H9N2 viruses, including I292V and V598I in the PB2 gene, Q226L in the HA gene, a 3 amino acid deletion (63–65) in the stalk region of NA, N30D, and T215A in the M1 gene, and P42S and a C-terminal truncation from 237 to 217 amino acids in the NS1 gene, etc. (Appendix A) [15,24,25,26,38,39,40,41,42,43,44,45,46]. The mutations in the 370 and 400 loops of 2SBS of NA tend to accumulate among H9N2 viruses, and SL20wt and SD1844 contained the highest numbers (Appendix A). The amino acid sequences of the HA and NA in SL20wt were 84.8% and 82.5%, identical to those of the 01310CE20. However, SL20wt had several different mutations in the HA, NA, and M2e regions that set it apart from the conventional vaccine strain 01310CE20 (Table 1). These mutations were located at epitope sites and were visible in the 3D structures of HA and NA (Appendix A). Unlike 01310CE20, SL20wt did not have N-glycosylation sites at 133–135 and 158–160 of HA and had different key amino acids in the M2e epitope, L10P, and T13N [11,47,48,49].

### 2.2. Recombinant Korean Y280-Lineage H9N2 Viruses Grow Well in ECEs Regardless of L226Q Mutation

Five recombinant viruses with different combinations of the wild-type or mutant HA (L226Q) and NA of SL20wt, and wild-type or mutant (MVV) PB2 of 01310 with six internal genomic segments of PR8, were generated. The replication efficiency in ECEs was compared with SL20wt (Table 2). The high virus titers of rSL20(P) and rSL20(P)-L226Q (10^9.41±0.12^ and 10^9.42±0.14^ EID_50_/mL, respectively) were comparable to that of SL20wt (10^9.67±0.29^ EID_50_/mL). The compatibility of SL20wt’s HA and NA genes with PR8’s six internal protein genes and the lack of effect of the L226Q mutation on virus replication in ECEs were demonstrated. However, the virus titer of rSL20(P)-310PB2 was significantly lower (10^8.57±0.46^ EID_50_/mL). To resolve this issue, the MVV mutations were introduced into 01310 PB2 (MVV310PB2) to increase polymerase activity, resulting in the increased virus titer of rSL20(P)-MVV310PB2 (10^9.58±0.14^ EID_50_/mL), comparable to SL20wt and others. Additionally, the MVV310PB2 gene was compatible with SL20wt’s internal protein genes, causing no significant decrease in the replication efficiency of rSL20-MVV310PB2 in ECEs (10^9.33±0.52^ EID_50_/mL).

### 2.3. MVV310PB2 Removes the Replication Capacity of PR8-Derived Recombinant Korean Y280-Lineage Virus in Mammalian Cells

rSL20(P) and rSL20(P)-L226Q had a higher ability to replicate and produce higher viral titers compared to the PR8 virus in both MDCK and A549 cells during 48 hpi. On the other hand, the SL20wt virus only replicated in MDCK but not in A549 cells (Figure 1). The replication of rSL20(P)-L226Q was similar to that of rSL20(P) in MDCK cells, but lower than that of rSL20(P) in A549 cells. The L226Q mutation had an effect on the replication efficiency only in A549 cells. The replication of rSL20(P)-310PB2 and rSL20(P)-MVV310PB2 was not successful in both MDCK and A549 cells due to the low activity of 01310 PB2 and MVV310PB2. However, rSL20-MVV310PB2 replicated in MDCK but not in A549 cells. The effect of MVV310PB2 may vary depending on the internal gene backgrounds of PR8 [rSL20(P)-MVV310PB2] and SL20wt [rSL20-MVV310PB2], and to determine this, the amino acid sequences of PB1 and PA from PR8 and SL20wt were compared. The results showed that L10 and L94 of MVV310PB2 may be in close proximity to the 691st and 576th PB1 residues of SL20wt (K691 and L576) and PR8 (R691 and I576), respectively. These differences are depicted in a 3D structure of the polymerase (Appendix A).

### 2.4. L226Q Mutation Increases Mammalian Pathogenicity of Recombinant Korean Y280-Lineage Virus

The pathogenicity of three virus strains, rSL20(P), rSL20(P)-L226Q, and rSL20(P)-MVV310PB2, was evaluated in BALB/c mice (Figure 2). The results showed that both rSL20(P) and rSL20(P)-L226Q caused a decrease in body weight, with rSL20(P)-L226Q leading to severe weight loss and ultimately euthanasia in all infected mice. The virus titer in the lungs was also 10 times higher for rSL20(P)-L226Q compared to rSL20(P). In contrast, rSL20(P)-MVV310PB2 did not cause any weight loss and was not re-isolated from the infected mouse lungs (Table 3).

### 2.5. The L226Q Mutation Reduces Susceptibility to Non-Specific Hemagglutination Inhibitors in Sera of SPF Chicken and Mouse

The susceptibility of rSL20(P) and rSL20(P)-L226Q to nonspecific inhibitors present in serum samples from SPF chickens and mice was compared using the hemagglutination inhibition (HI) test (Figure 3). The results showed that rSL20(P) was more susceptible to the inhibitors compared to rSL20(P)-L226Q. The virus produced relatively large, clear dot-shaped, unagglutinated RBC precipitates in the middle of the well bottom. Treatment of the chicken and mouse serum samples with RDE and heat eliminated the inhibitory effects on both viruses, leading to a reduction of the unagglutinated RBC precipitates in the well bottom.

### 2.6. Monovalent Inactivated Oil Emulsion Vaccines Induce Very Low HI Antibody Titers to Heterogeneous Antigens

Three inactivated oil emulsion vaccines, V-rSL20(P), V-rSL20(P)-L226Q, and V-rSL20(P)-MVV310PB2, were inoculated into five three-week-old SPF chickens and the serum collected at three and four weeks post vaccination (wpv) for evaluation of their immunogenicity. They elicited high HI titers with a minimum of 256 at 3 wpv (Table 4). The vaccine with the highest mean HI titer was V-rSL20(P)-MVV310PB2, despite having a slightly lower virus titer than V-rSL20(P). The vaccine with the lowest HI titer was V-rSL20(P)-L226Q, regardless of whether the antiserum sample was heterologous or homologous. This suggests that the L226Q mutation may affect the antigenic structure and the binding of antibodies to HA epitopes.

### 2.7. Bivalent Inactivated Oil Emulsion Vaccine Composed of Y439/Korea- and Korean Y280-Lineage H9N2 Viruses Compensates Skewed Humoral Immunity of Monovalent Vaccines

The immunogenicity of monovalent and bivalent oil emulsion vaccines composed of r310-NS28 and rSL20(P)-MVV310PB2 of different antigenic structures was compared (Table 5). The r310-NS28 is a recombinant 01310CE20 strain which is more productive and less pathogenic than the commercial vaccine strain (01310CE20) due to replaced NS genome of another H9N2 strain (KBNP0028) [25]. The monovalent V-r310-NS28 vaccine, with an average of 10^8.75^ (Exp. 1) and 10^9.25^ EID_50_/mL of r310-NS28 (Exp. 2), induced 128 and 294 HI titers at 3 wpv and 294 and 294 HI titers at 4 wpv to r310-NS28, respectively. However, it induced lower HI titers to rSL20(P)-MVV310PB2. The monovalent V-rSL20(P)-MVV310PB2 vaccine, with an average of 10^9.00^ and 10^9.00^ EID_50_/mL of rSL20(P)-MVV310PB2 (Exp. 1 and 2), induced 2048 and 891 HI titers at 3 wpv and 3104 and 446 HI titers at 4 wpv in average to rSL20(P)-MVV310PB2, respectively, but did not induce any detectable HI titers to r310-NS28. The bivalent vaccines, composed of 10^8.75^ EID_50_/mL of r310-NS28 and 10^9.00^ EID_50_/mL of rSL20(P)-MVV310PB2 (Exp. 1), induced higher HI titers at 3 wpv (338 vs. 128 to r310-NS28 and 2353 vs. 2048 to rSL20(P)-MVV310PB2) and lower HI titers at 4 wpv (194 vs. 294 to r310-NS28 and 1783 vs. 3104 to rSL20(P)-MVV310PB2) compared to the corresponding HI titers of the monovalent vaccines. However, the bivalent vaccine composed of 10^9.25^ r310-NS28 and 10^9.00^ EID_50_/mL rSL20(P)-MVV310PB2 (Exp. 2) induced lower HI titers at 2 (11 vs. 74 to r310-NS28 and 169 vs. 194 to rSL20(P)-MVV310PB2), 3 (194 vs. 294 to r310-NS28 and 388 vs. 891 to rSL20(P)-MVV310PB2), and 4 wpv (147 vs. 294 to r310-NS28 and 223 vs. 446 to rSL20(P)-MVV310PB2) compared to the corresponding HI titers of monovalent vaccines. As the bivalent vaccine induced the antibody to both antigens, it is better than the monovalent vaccines. The bivalent vaccine composed of 10^8.75^ of r310-NS28 and 10^9.00^ of EID_50_/mL rSL20(P)-MVV310PB2 is more preferable than the bivalent vaccine composed of 10^9.25^ r310-NS28 and 10^9.00^ EID_50_/mL rSL20(P)-MVV310PB2 due to comparable antibody titers to the corresponding monovalent vaccines.

### 2.8. Comparison of Specific Neuraminidase Inhibiting Activities of Serum Samples from Monovalent and Bivalent Vaccine-Inoculated Chickens

The NA activities of rSL20(P)-MVV310PB2 and r310-NS28 were similar when tested at the same HA titers (Figure 4A). However, to accurately compare the neuraminidase-inhibiting (NI) activities, the nonspecific neuraminidase inhibitors needed to be removed by treatment with RDE and subsequent heating for 8 h to remove residual RDE activity (Figure 4B) [50]. Only the heat-treated serum samples from Exp. 2 of the monovalent and bivalent vaccine experiments were used to compare the NI activities between monovalent- and bivalent-vaccine-inoculated chickens. The serum samples from the monovalent vaccine groups (V-rSL20(P)-MVV310PB2 and V-r310-NS28) showed significantly higher NI activity against the homologous antigens and insignificantly lower activity against the heterologous antigens compared to the negative control groups. The serum samples from the bivalent vaccine group showed significantly higher, but insignificantly lower, NI activity against both rSL20(P)-MVV310PB2 and r310-NS28 compared to the control groups (Figure 4C,D). This suggests that the bivalent vaccine may not effectively induce NI antibodies against the NA of r310-NS28.

## 3. Discussion

Nation-wide recurrent outbreaks of the Y439/Korea-lineage virus in poultry since 1999 might have started with the multi-focal introduction of similar but apparently different viruses from migratory wild birds. The outbreaks might have resulted from farm-to-farm secondary transmission [26,51]. The fact that there have been no reports of isolation of new genotypes in vaccinated layer farms reflects the low possibility of reverse flow of viruses from LPMs and the presence of an unknown route of virus transmission from migratory wild birds to layer farms. The recent isolation of another new Y439/Korea-lineage virus from a migratory wild bird may indicate a continuous, unignorable virus load in the environment and the potential risk of poultry farm transmission, as before [9,52]. The relatively high variability of HA (0.71%) and NA (0.64%) between Korean Y280-lineage viruses in 2020, given the error rate of the polymerase and the absence of humoral immunity pressure, supports the hypothesis of multi-focal infections of similar but not directly related viruses during the time period [53]. The route and method of transmission of chicken-adapted viruses from China to Korea is mysterious because similar viruses have never been reported in migratory wild birds under intensive virus monitoring [9].

The distribution of Korean Y280-lineage viruses in infected chicken tissues differs from that of Y439/Korea-lineage viruses, indicating differences in pathogenicity and level of chicken adaptation [9,10]. The preference for oropharyngeal virus shedding to the cloaca and rapid airborne transmission to contact chickens seen in Korean Y280-lineage viruses, as well as their replicability in extra-pulmonary tissues such as the brain, thymus, and spleen, may be related to differences in mutations, especially in HA and NA (Appendix A) [54,55,56]. In contrast to early K1 genotype viruses, such as 01310CE20 and KBNP0028, the high virus titer of SL20wt in ECEs without passages reflects its chicken adaptation [4,5]. The comparable virus titers of rSL20(P) and rSL20(P)-L226Q with that of SL20wt indicate the optimal compatibility of PR8’s internal proteins with SL20wt’s HA and NA, with or without the L226Q mutation. The increase in virus titer seen upon the introduction of MVV mutation into 01310 PB2 highlights the need to increase the polymerase activity of the trimer composed of 01310 PB2, PR8 PB1, and PA, which can be modulated only by PB2 to balance its activity with that of SL20wt HA/NA [15,28]. The compatibility of MVV310PB2 with the proteins which are expressed from the seven genomes of SL20wt is also demonstrated by the high virus titer of rSL20-MVV310PB2.

Despite similarities in growth kinetics between SL20wt and PR8 in MDCK cells, the public health risk of SL20wt may not be high as it does not replicate in A549 cells. However, the higher virus titers of rSL20(P) and rSL20(P)-L226Q than rPR8 in both cells raises strong public health concerns about the conventional reverse genetics system using the highly pathogenic PR8 PB2 gene. Our experiences in testing the growth kinetics of various PR8-derived clade 2.3.2.1c H5N1, clade 2.3.4.4 H5N6 and H5N8, Y439/Korea-lineage, and G1-lineage Egyptian H9N2 recombinant strains in MDCK and/or A549 cells showed that rSL20(P) and rSL20(P)-L226Q outgrow rPR8, similar to the G1-lineage Egyptian H9N2 strain [25,30,35,36]. However, the similar growth kinetics of rSL20-MVV310PB2 to rPR8 in MDCK cells was unexpected, as MVV310PB2 did not have a dominant-negative effect. This may be due to differences in the interaction of MVV310PB2 with PB1 and PA of SL20wt and PR8, as evident from a comparison of PB1 and PA amino acid sequences of SL20wt, 01310 E20, and PR8. We found different amino acid residues of PB1 (I576L and R691K) in SL20wt compared to those of PR8 in the PB2 and PB1 interfaces (Appendix A). Additionally, recent H3N2 seasonal flu viruses have acquired the R691K mutation in PB1. Further experimental data are needed to conclude that the better interaction of MVV310PB2 with SL20wt PB1 may increase polymerase activity enough to overcome the dominant-negative effect of MVV310PB2 and replicate in MDCK [57].

The L226Q mutation decreased the replication efficiency in A549 cells but increased its pathogenicity in BALB/c mice. This resulted in severe weight loss and a 10-fold higher virus titer of rSL20(P)-L226Q in the lungs compared to rSL20(P) (Figure 1 and Figure 2, and Table 3). The conflicting results may be partially explained by the increased resistance of rSL20(P)-L226Q to non-specific inhibitors in SPF chicken and mouse serum. A similar result was also reported in the G1-lineage Egyptian H9N2 strains (Figure 3) [35]. Serum contains heat-stable α and γ inhibitors, as well as heat-labile β inhibitors. The sialic acid in α and γ inhibitors interacts with the receptor binding site of HA, while the mannose-binding lectin activity in β inhibitors interacts with mannose-rich glycans on the HA and NA [58,59,60]. RDE and heat treatment remove the inhibitory activity of serum, which may contain sialic acid [35,61,62]. The increased affinity of L226Q to the avian receptor may increase its susceptibility to non-specific inhibitors in serum. However, the decreased susceptibility of L226Q can only be explained by its increased coordination with NA to remove the activity of non-specific inhibitors. In a previous study, we observed lower immunogenicity in the G1-lineage Egyptian H9N2 vaccine strain with the L226Q mutation. This was hypothesized to be due to the masking of HA epitopes by formalin-mediated cross-linking of bound sialoproteins [35]. In this study, we did not find an effect of L226Q on immunogenicity, but the HI titer to rSL20(P)-L226Q was significantly lower than that of rSL20(P) (Table 4). This suggests that L226Q directly or indirectly changes the antigenic structure of HA. Therefore, the reverse mutation (L226Q) to reduce the mammalian pathogenicity of PR8-derived H9N2 vaccine strains is not recommended.

The optimal vaccine, V-rSL20(P)-MVV310PB2, induced very high HI antibody titers to the homologous antigen, but not to the heterologous antigen (r310-NS28). In contrast, the V-r310-NS28 induced low but cross-reactive HI antibodies to rSL20(P)-MVV310PB2 (Table 5). The result may be due to differences in the glycosylation of HA epitopes (Table 1). The N-glycans of r310-NS28 at 133N and 158N may mask major epitopes of HA, but the unmasked other epitopes may be the target of humoral immunity. Conversely, the unglycosylated epitopes of rSL20(P)-MVV310PB2 may be the major target of humoral immunity, but the antibody cannot bind to the epitopes of r310-NS28 due to the steric hindrance of the N-glycans. The result supports the importance of the epitopes and the preferred acquisition of 144N or 158N glycans to evade vaccine-induced antibodies by clade 2.3.2.1c H5N1 viruses [63]. A limitation of our study was that the inactivated virus antigens were not assayed by HAU before the vaccine was administered as a vaccine. This means that the chickens may have received different doses of inactivated virus and this could be the reason for the differences observed in HI titers in which case the differences in HI titers may not be due to inherent differences in antigenicity between the vaccine candidates. In other studies, it is typical to first titer inactivated virus and normalizing HA units before using them as vaccine. HAUs are thus the units used for the vaccine dose. The NI test result is affected by non-specific α, β, and γ inhibitors and usually requires RDE and proper heat treatments [50]. In this study, we found that heat treatment at 56 °C for 30 min after RDE treatment was not enough to remove the neuraminidase activity, which interfered with the NI test. The heat treatment may not have removed the heat-stable inhibitors in the tested serum samples. However, the serum samples from the vaccinated groups exhibited significantly higher NI activities compared to the negative control (Figure 4). The reason why the bivalent vaccine did not induce NI antibodies to r310-0028 is unclear, but the short NA stalk (18 amino acid deletion) of r310-0028 may decrease immunogenicity in the presence of the intact NA of rSL20(P)-MVV310PB2 [64]. Although the bivalent vaccine has less immunogenicity to the NA of r310-0028, the very low HI titers of the monovalent vaccines to heterologous antigens may encourage bivalent vaccines to show better HI titers to both antigens if Y439/Korea-lineage viruses continue to be isolated in migratory wild birds.

## 4. Materials and Methods

### 4.1. Viruses, Plasmids, Cells, and Eggs

The A/chicken/Korea/SL20/2020 (H9N2) (SL20) was isolated from oropharyngeal swab samples of Korean native chickens at a LPM of Korea and cultured in specific-pathogen-free (SPF) ECEs (VALO Biomedia GmbH, Osterholz-Scharmbeck, Germany). The previously developed Y439/Korea-lineage H9N2 recombinant r310-NS28 vaccine strain was used for monovalent and bivalent vaccine preparation [25]. Recombinant viruses were generated by a pHW2000 plasmid-based reverse genetics system provided by Dr. Robert Webster (St. Jude Children’s Research Hospital, Seattle, WA, USA). Six internal genomic segments (PB1, PB2, PA, NP, M, and NS) of A/Puerto Rico/8/1934 (H1N1) (PR8), and PB2 and mutated PB2(MVV) of A/chicken/Korea/01310/2001 (H9N2) (01310CE20), were used as in a previous report. The eight genome segments of the SL20 virus were amplified using universal primer sets and cloned into the pHW2000 vector [15,65]. The 293T, MDCK, and A549 cell lines (KCTC, Daejeon, Republic of Korea) were maintained in DMEM supplemented with 10% FBS, or in DMEM/F12 supplemented with 10% FBS, respectively (Life Technologies Co., Calsbad, CA, USA). The recombinant virus was produced in 293T cells as described previously and then propagated twice in 10-day-old SPF ECEs prior to experimentation [66].

### 4.2. RT-PCR, Sequencing, and Sequence Analysis

RNA was extracted from harvested allantoic fluid using a Viral Gene-spin™ Viral DNA/RNA Extraction Kit (iNtRON Biotechnology, Seongnam, Gyeonggi, Republic of Korea) and cDNA was synthesized by a TOPscript™ cDNA Synthesis kit (Enzynomics, Daejeon, Republic of Korea). Full genomes were amplified using universal primer sets and sequenced as previously described [51,65]. The nucleotide and amino acid sequences of SL20 were compared with other genes in the Genbank using BLAST. The BioEdit program (v7.2.5) was used for nucleotide sequence translation, amino acid comparison, and calculation of amino acid identity. The variable amino acids of HA and NA were located in the 3D structures using PyMOL program and amino acids closely located in the interfaces of polymerase was visualized using 3D view of RCSB PDB “https://www.rcsb.org/3d-view (accessed on 12 January 2023).

### 4.3. Generation of Recombinant Viruses by Site-Directed Mutagenesis and Reverse Genetics

The L226Q mutation was introduced into the HA gene of SL20 using the Muta-direct site-directed mutagenesis kit (iNtRON, Seongnam, Republic of Korea) with the specific primer sets detailed in Appendix A following the manufacturer’s protocol. Additionally, the PB2 gene of the 01310 virus, which had I66M, I109V, and I133V mutations, was used in the recombinant virus [15,28]. The pHW2000 vectors containing each genome segment were sequenced using cmv-SF and bGH-SR primer sets listed in Appendix A.

The generation of various recombinant Y280-lineage H9N2 viruses was carried out to obtain different gene constellations and mutations as listed in Table 2. The eight plasmids for each genome segment were mixed in Opti-MEM (Life Technologies) at a concentration of 300 ng per plasmid. The mixture was then transfected into confluent 293T cells in a six-well plate using the Plus reagent and Lipofectamine 2000 (Life Technologies). After overnight incubation, 1 µL of Opti-MEM and L-1-tosylamido-2-phenylethyl chloromethyl ketone (TPCK)-treated trypsin (0.5 mg/mL) (Sigma-Aldrich, St. Louis, MO, USA) were added to each well. The supernatant was harvested 24 h later and 200 μL of the supernatant was inoculated into the allantoic cavity of 10-day-old SPF ECEs to propagate the virus for 72 h at 37 °C. The presence of the virus was confirmed by hemagglutination assay using 1% (*v*/*v*) chicken red blood cells, and the full genomes of the virus were confirmed by RT-PCR and Sanger sequencing [51]. The confirmed virus was passaged once, and the second passaged virus (CE2) was stored at −80 °C until use.

### 4.4. Titration of Recombinant Viruses in ECEs

The titer of each virus was determined as the 50% egg infectious dose (EID_50_) by inoculating serially 10-fold diluted virus into 5 10-day-old SPF ECEs. After incubating the virus-inoculated eggs at 37 °C for 72 h, followed by chilling at 4 °C before harvest, the EID_50_ was calculated using the hemagglutination test in each dilution and the Spearman–Karber method [67]. The replication efficiency of each virus was compared by EID_50_ of harvested virus after inoculating 100 EID_50_/0.1 mL of each virus into 10-day-old SPF ECEs and incubating for 72 h.

### 4.5. Growth Kinetics and Pathogenicity of Recombinant Viruses in Mammalian Hosts

Virus replication in mammalian cells, specifically MDCK and A549 cells, and the pathogenicity of the viruses in mice were evaluated in order to select a safe recombinant Y280-lineage H9N2 vaccine strain that was free from potential human infection risks. The viruses were diluted in DMEM with 1 µg/mL TPCK-trypsin and 0.1 MOI of virus was infected with confluent MDCK and A549 cells in a 12-well plate. Following a 1-h incubation period, the inoculum was replaced with fresh media, and the supernatant was harvested at specific time points (0, 24, 48, and 72 h) to measure the viral titer using the 50% tissue culture infectious dose (TCID_50_) in MDCK cells. The supernatant was serially diluted and inoculated into MDCK cells in a 96-well plate, and the virus in each dilution was checked by hemagglutination assay after 72 h to calculate the TCID_50_ using the Spearman–Karber method.

The mouse infection experiment was approved by the Institutional Animal Care and Use Committee (IACUC) of Seoul National University (IACUC-SNU-210811-2) and conducted in an animal biosafety level 2 facility at the Animal Center for Pharmaceutical Research of Seoul National University (Seoul, Republic of Korea) according to the national guidelines for the care and use of laboratory animals. Eight six-week-old female BALB/c mice (KOATEC, Pyeongtaek, Republic of Korea) of each group were sedated by an intraperitoneal injection of Zoletil 50 (15 mg/kg; Virbac, Carros, France), and 10^6^ EID_50_/0.1 mL of each recombinant virus was inoculated via the intranasal route. The negative control group (mock) was inoculated with the same volume of sterile phosphate-buffered saline (PBS). Three of eight mice in each group were euthanized through CO_2_ asphyxiation to test virus replication and virus titers in the lungs at three days-post-inoculation (dpi). The remaining mice were observed for weight loss and mortality for 7 dpi. The mice were euthanized by CO_2_ asphyxiation when the body weight of each mouse was reduced by more than 20%. The collected lungs were homogenized using a TissueLyzer 2 (Qiagen, Valencia, CA, USA) equipped with 5 mm diameter stainless steel beads and mixed with PBS (10 volumes of lung weight). Following centrifugation at 13,000 rpm for 10 min, 0.1 mL of each supernatant and 10-fold diluted pooled supernatants were inoculated into ECEs as described above. The presence of virus in harvested allantoic fluid was tested by the plate HA test with 1% chicken RBCs. The virus titer of pooled lung specimens was calculated as described above.

### 4.6. Susceptibility Test of Recombinant Viruses to Non-Specific Inhibitors in SPF Chicken and Mouse Sera

Serum samples from SPF chickens and BALB/c mice were prepared to detect non-specific inhibitors of the viral hemagglutination. One group of serum was heat-treated at 56 °C for 30 min, while the other group was treated with three volumes of RDE (Receptor Destroying Enzyme II, Denka Seiken Co., Ltd., Tokyo, Japan) at 37 °C overnight before heat inactivation at 56 °C for 30 min. The hemagglutination inhibition (HI) test was performed as below. The serum was serially 2-fold diluted with PBS, and 25 µL of the diluted serum was mixed with the same volume of the antigen virus with 4 hemagglutination unit. They were incubated at 4 °C for 40 min to allow binding of any non-specific inhibitors present in the serum before the hemagglutination reaction of the virus. Then, 25 µL of 1% (*v*/*v*) chicken RBCs were added and incubated for another 40 min at 4 °C, and the HI result was recorded [68]

### 4.7. Efficacy Test of Monovalent and Bivalent Inactivated Oil Emulsion Vaccines in Chickens

Each virus (100 EID_50_/100 µL) was inoculated into SPF ECEs for propagation and titrated as EID_50_/mL prior to inactivation. The recombinant virus was then inactivated using 0.2% formaldehyde (Sigma-Aldrich) and inoculated into 10-day-old SPF ECEs after a 24-h incubation period to confirm virus inactivation. The monovalent and bivalent vaccines were prepared by mixing one or two inactivated viruses with ISA70 (SEPPIC, Courbevoie, France) in a ratio of 3:7 (virus-to-oil). A negative control vaccine was prepared by utilizing negative allantoic fluid of 10-day-old ECEs as above.

The animal experiment was approved by the Institutional Animal Care and Use Committee (IACUC) of Seoul National University (IACUC-SNU-220806-1) and conducted in an animal biosafety level 1 facility of Seoul National University. Five three-week-old SPF chickens per each vaccination group were subcutaneously vaccinated with 0.5 mL of the prepared inactivated oil emulsion vaccine. Serum samples were collected at 0, 3, and 4 weeks post vaccination (wpv), and serum antibody titers were determined using the HI test [68]

The neuraminidase activity of virus and neuraminidase inhibiting activity of serum samples were measured using the NA-star™ Influenza Neuraminidase Inhibitor Resistance Detection Kit. (Thermo Fisher Scientific, Waltham, MA, USA) according to the manufacturer’s manual. The NA activity of virus with varying HA titers ranging from 21 to 27 was measured and 8 HAU of virus was selected for NI test. The heat-treated (56 °C for 30 min) serum samples collected at 3 wpv of each group in Exp. 2 was utilized to compare the NI antibody titers. Briefly, serially 2-fold diluted serum sample was incubated with the same volume of virus (25 µL:25 µL) at room temperature for 20 min. A total of 10 µL of 1:1000 diluted NA substrate was added to every well and incubated at room temperature for 30 min. Then, 60 µL of NA-star accelerator was added, and luminescence was measured immediately using an Infinite 200 PRO (TECAN, Männedorf, Switzerland).

### 4.8. Statistical Analysis

The significance of the results between experimental groups was evaluated by one-way analysis of variance (95% confidence intervals) (IBM SPSS statistics, Armonk, NY, USA), and *p* < 0.05 was defined as statistically significant.

## 5. Conclusions

The optimal recombinant vaccine strains for AIVs can be engineered by balancing the activity of HA/NA with polymerase activity, and the polymerase activity can be fine-tuned through PB2 mutations. To protect against antigenically distinct H9N2 viruses, a bivalent vaccine is better than a monovalent vaccine, but the ratio of antigens in the bivalent vaccine needs to be optimized for improved vaccine efficacy.

## 6. Patents

Korea pending patent 10-2021-0191058 (MAMMALIAN AVIRULENT AND EMBRYONATED EGG HIGHLY REPLICATIVE RECOMBINANT INFLUENZA VIRUS).

## Figures and Tables

**Figure 1 ijms-24-08840-f001:**
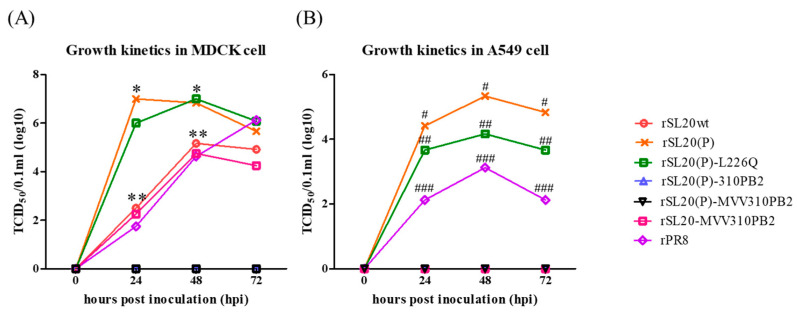
Growth kinetics of recombinant and wild-type Korean Y280-lineage H9N2 viruses. Recombinant SL20 viruses and wild-type SL20 virus (SL20wt) (0.1 MOI) were infected into (**A**) MDCK cells and (**B**) A549 cells. After 1h incubation, inoculum was replaced with fresh media and supernatant was obtained at each time point (0, 24, 48, and 72 h). The viral titer was measured as TCID_50_/mL in MDCK cells, and the result was the average of three independent repeated experiments. Statistical significance was analyzed by one-way ANOVA and significant difference with other groups was marked in panels (**A**) *, rSL20(P) and rSL20(P)-L226Q; **, SL20wt and rSL20-MVV310PB2, and (**B**) #; rSL20(P); ##; rSL20(P)-L226Q; ###; rPR8 (*p* < 0.05).

**Figure 2 ijms-24-08840-f002:**
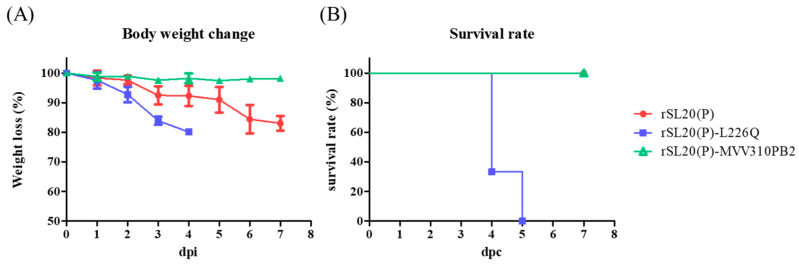
Weight loss and survival rate of recombinant Korean Y280-lineage H9N2 virus-infected mouse. Five six-week-old female BALB/c mice in each group were infected with 10^6^ EID_50_/50 μL via intranasal route and weighed over the course of 1 week. Mice with a weight loss of 20% or more were considered dead and euthanized, and average weight loss (**A**) and survival rate of each group (**B**) are shown.

**Figure 3 ijms-24-08840-f003:**
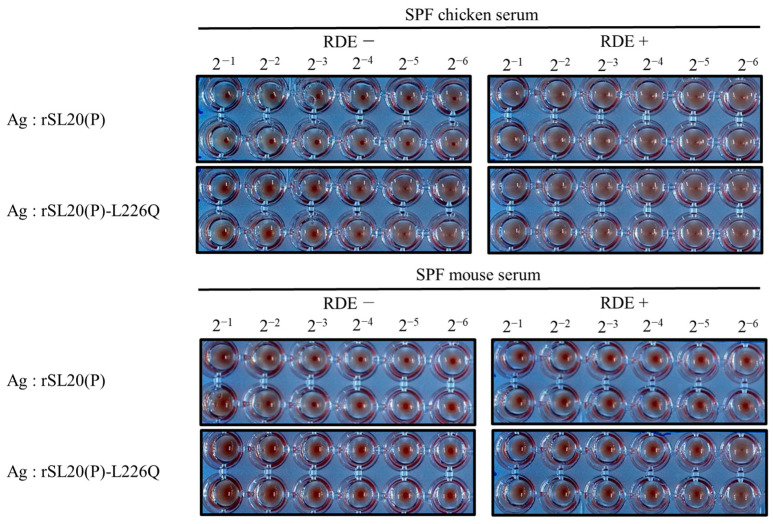
Hemagglutination inhibition test of SPF chicken and mouse serum samples against recombinant SL20 viruses. Serum samples from SPF chickens and BALB/c mice were treated with two methods: RDE− with heat-treated serum only (56 °C, 30 min) and RDE+ with RDE and heat-treated serum for inactivation of non-specific inhibitors. Each serum sample was serially two-fold diluted with PBS and reacted with the same volume of virus (4 HAU). After 40 min incubation at 4 °C, 1% chicken RBC was added, and the results were recorded after 40 min at 4 °C.

**Figure 4 ijms-24-08840-f004:**
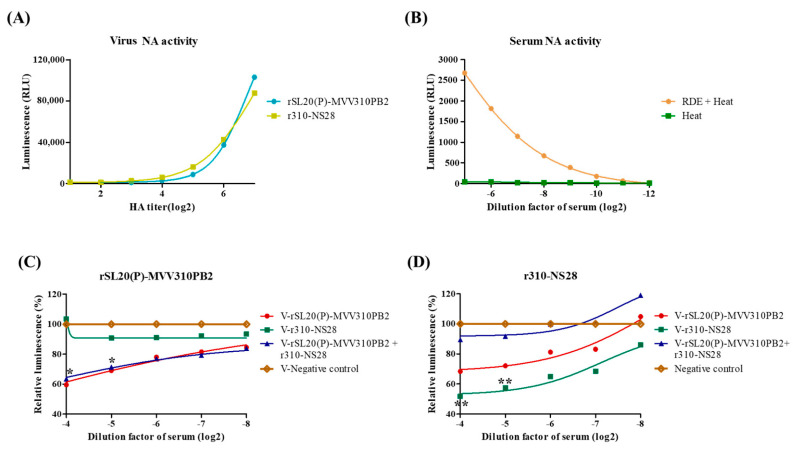
Neuraminidase-inhibiting activity of serum samples from vaccinated chickens. Neuraminidase activities of (**A**) H9N2 vaccine strains, rSL20(P)-MVV310PB2, and r310-NS28; (**B**) differently treated SPF chicken serum, RDE+ heat treatment (56 °C, 30 min), and heat treatment. Neuraminidase-inhibiting activity of heat-treated serum samples from vaccinated and negative control groups of Exp. 2. Serum samples collected at 3 wpv were reacted with 8 HAU of (**C**) rSL20-MVV310PB2 and (**D**) r310-NS28. The data are shown as the relative neuraminidase activity (%) of the average of the five chicken serums in each vaccine group to that of the negative control group. *, rSL20(P)-MVV310PB2 and bivalent vaccine groups were significantly different from V-r310-NS28 and negative control; **, V-r310-NS28 group was significantly different from the bivalent vaccine and negative control groups.

**Table 1 ijms-24-08840-t001:** Variable amino acid residues in the epitopes of HA, NA, and M2e.

HA				NA		
H3 Numbering	H9 Numbering	SL20wt	01310 CE20	SL20 Numbering	SL20wt	01310 CE20
96	107	L	M	125 (122)	G	S
	133	Q	L	126 ^b^	L	S
	138	T	R	127 ^b^	G	N
133	145	S	N (NG) ^a^	187 ^b^	K	R
145	153	D	G	199 (196)	K	R
156	164	Q	H	248 (245)	G	G
158	166	N	N (NG) ^a^	249 ^b^	K	R
160	168	A	S	253 (250)	R	R
183	191	N	H	296 (293)	K	K
189	197	D	T	344 (341)	R	R
190	198	V	E	346 ^b^	A	N
192	200	T	M	356 (353)	N	Y
205	213	A	T	360 ^b^	I	V
208	216	E	D	367 ^b^	K	S
216	224	L	V	368 (365)	E	K
226	234	L	Q	400 (397)	S	N
273	282	K	N	401 ^b^	D	N
274	283	M	S			
275	285	S	N	M2e		
276	287	T	I	numbering	SL20	1310CE20
306	315	S	P	10	L	P
325	334	S	A	13	T	N

^a^ N-linked glycosylation site. ^b^ Variable amino acid neighboring reported epitope.

**Table 2 ijms-24-08840-t002:** Virus titers of SL20wt and recombinant SL20 viruses in ECEs.

Virus	HA	NA	PB2	PB1	PA	NP	M	NS	EID_50_/mL(log10) ^a^
SL20wt	SL20	SL20	SL20	SL20	SL20	SL20	SL20	SL20	9.67 ± 0.29
rSL20(P)	SL20	SL20	PR8	PR8	PR8	PR8	PR8	PR8	9.41 ± 0.12
rSL20(P)-L226Q	SL20-L226Q	SL20	PR8	PR8	PR8	PR8	PR8	PR8	9.42 ± 0.14
rSL20(P)-310PB2	SL20	SL20	01310	PR8	PR8	PR8	PR8	PR8	8.57 ± 0.46 *
rSL20(P)-MVV310PB2	SL20	SL20	01310-MVV ^b^	PR8	PR8	PR8	PR8	PR8	9.58 ± 0.14
rSL20-MVV310PB2	SL20	SL20	01310-MVV	SL20	SL20	SL20	SL20	SL20	9.33 ± 0.52

^a^ EID_50_/mL, 50% chicken embryo infection dose. ^b^ 01310 PB2 gene with I66M, I109V, and I133V mutations. * Significantly different from others (*p* < 0.05).

**Table 3 ijms-24-08840-t003:** Virus isolation rates and virus titers of the lungs of recombinant Korean Y280-lineage H9N2 virus-infected mice.

Recombinant Virus	Virus Isolation Rate	TCID_50_/0.1 mL ^a^ (log10)
rSL20(P)	3/3	4.00 ± 0.25
rSL20(P)-L226Q	3/3	5.00 ± 0.25
rSL20(P)-MVV310PB2	0/3	0.00 ± 0.00

^a^ TCID_50_/0.1 mL, 50% tissue culture infection dose.

**Table 4 ijms-24-08840-t004:** Serum HI antibody titers of monovalent vaccines.

Vaccine	Dose (EID_50_/mL, log10)	Weeks Post Vaccination (wpv)	HI Titer (GMT ^a^)		
			rSL20(P)	rSL20(P)-L226Q	rSL20(P)-MVV310PB2
V-rSL20(P)	9.25	3 wpv	1195 (756–1633)	256 (256–256)	1195 (756–1633)
		4 wpv	2303 (721–3884)	288 (90–486)	1195 (461–1929)
V-rSL20(P)- L226Q	8.75	3 wpv	1741 (888–2594)	256 (61–451)	1075 (283–1867)
		4 wpv	1638 (942–2355)	307 (66–548)	1331 (478–2184)
V-rSL20(P)- MVV310PB2	9.00	3 wpv	2560 (653–4467)	371 (117–625)	2304 (125–4483)
		4 wpv	2048 (491–3605)	666 (239–1092)	1434 (737–2130)
Negative control (allantoic fluid)	-	3 wpv	<2	<2	<2
		4 wpv	<2	<2	<2

^a^ Geometric mean HI titer with 95% confidence interval.

**Table 5 ijms-24-08840-t005:** Comparison of HI antibody titers of monovalent and bivalent vaccines.

Vaccine	Dose (EID_50_/mL, log10)		Weeks Post Vaccination (wpv)	GMT of HI Titer ^a^			
	Exp. 1	Exp. 2					
				r310-NS28		rSL20(P)-MVV310PB2	
				Exp. 1	Exp. 2	Exp. 1	Exp. 2
V-r310-NS28	8.75	9.25	2 wpv	NT	74 (29–226)	NT ^b^	9 (3–33)
			3 wpv	128 (23–716)	294 ^†^ (143–604)	111 (14–869)	21 (7–67)
			4 wpv	294 (41–2092)	294 ^†^ (143–604)	55 (7–434)	42 (13–134)
V-rSL20(P)-MVV310PB2	9.00	9.00	2 wpv	NT	<2	NT	294 (143–604)
			3 wpv	<2	<2	2048 ^‡^ (866–4842)	891 ^‡^ (434–1831)
			4 wpv	<2	<2	3104 ^‡^ (1938–4973)	446 (217–916)
V-r310-NS28 + rSL20(P)-MVV310PB2	8.75 + 9.00	9.25 + 9.00	2 wpv	NT	10.6 (4–28)	NT	169 (35–806)
			3 wpv	338 ^†^ (156–729)	194.0 (90–419)	2353 ^‡^ (1601–3457)	388 (180–838)
			4 wpv	194 (72–517)	147.0 (72–302)	1783 (867–3663)	223 (87–572)
Negative control (allantoic fluid)	-	-	2 wpv	<2	<2	<2	<2
			3 wpv	<2	<2	<2	<2
			4 wpv	<2	<2	<2	<2

^a^ Geometric mean HI titer with 95% confidence interval. ^b^ NT: not tested. ^†^ Significantly different from corresponding samples of V-rSL20(P)-MVV310PB2 (*p* < 0.05). ^‡^ Significantly different from corresponding samples of V-r310-NS28 (*p* < 0.05).

## Data Availability

Not applicable.

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
