# Peer review of "Engineering an Optimal Y280-Lineage H9N2 Vaccine Strain by Tuning PB2 Activity"

_ijms, 2023, doi:10.3390/ijms24108840_

Round 1
Reviewer 1 Report
Summary
The paper by An et al entitled, “Engineering optimal Y280-lineage H9N2 vaccine strain by tuning PB2 activity” describes a study aiming to create vaccine candidate/s that has a better safety profile and remains efficacious. Here, the investigators rescued several vaccine viruses using the PR8 and SL20 genome segments. Replication of these viruses were assayed in mammalian in vitro and in vivo systems to examine replication potential in mammalian hosts. Antigenicity of inactivated vaccine candidates was then examined in chickens. Other assays such as HI and NI tests were also performed to further characterize the vaccine viruses.
Major comments
Overall: Please improve upon the organization and flow of the paper. It is important to give the reader context and provide rationale on the experimental setup. As it is written, the results are presented as unrelated sections and it is not obvious to me as to why an experiment was performed. Also, there are key information in the methods section that are needed to understand the results. Going from one section to another can be distracting for the reader. Lastly, there are many abbreviations and names that are not defined clearly, and some names are complicated and hard for a reader to remember.
Line 230-231: Before describing the results, please briefly describe the vaccination protocol used to obtain the sera used for HI assays.
Line 246-271, Table 4: Have you done any statistical tests to determine differences in HI titers? Also, L226Q tends to increase avidity to RBCs, how would that affect HI titer results?
Line 329-330: I think that this statement is inaccurate. Since the MPMs are associated with pathogenicity in mammals, it is unclear as to whether it contributes to changes in tropism in chickens.
Line 526-533: Can you provide an explanation as to how the dose for the inactivated viruses was determined? It appears that the viruses were not assayed (e.g. by HA) before it was administered as a vaccine. This means that the chickens received different doses of inactivated virus and thus, can be the reason for the differences observed in HI titers. If this is the case, the differences in HI titers would not be due to inherent differences in antigenicity between the vaccine candidates. In other studies, it is typical to first titer inactivated virus and normalizing HA units before using them as vaccine. HAUs are thus the units used for the vaccine dose.
Minor comments
Line 73-77: While these are known MPMs, viruses isolated from patients infected with avian influenza often retain avian-like markers, suggesting that these MPMs are not absolutely necessary for a virus to infect mammals. Additionally, these MPMs often need a particular genetic background to “work” as an MPM. In other words, genetic context matters for mutations to confer phenotypic change.
Figure 2A. Please indicate error bars for the average weights
Lines 101-102: I think it would be more useful to define what “poor results” refers to. E.g., does it refer to reduced virus shedding and/or clinical signs in vaccinated birds and so on.
Line 114: Please define MVV. Additionally, also avoid too many abbreviations to make the text more readable.
Line 123: Please provide a brief history of the SL20wt isolate. Where and which host species was it isolated from? Any clinical information regarding the bird from which SL20wt was isolated from?
Line 134-126: Why was SL20 sequence compared to that of N20-99 and SD1877? Is it because these are prototypical H9N2 viruses that are currently circulating in China and Korea?
Line 131: Is the stalk region here of the neuraminidase protein?
Line 279: Please provide a rationale in using r310-NS28 virus in this experiment.
Line 599: Please fill out the section on Conflict of interest.
Misc: What is the subtype of r310-NS28? If it is different, it can help explain the NI results.
Author Response
Response to reviewer’s comments
Thank you for the precious comments and our manuscript improved apparently.
Reviewer 1
Summary
The paper by An et al entitled, “Engineering optimal Y280-lineage H9N2 vaccine strain by tuning PB2 activity” describes a study aiming to create vaccine candidate/s that has a better safety profile and remains efficacious. Here, the investigators rescued several vaccine viruses using the PR8 and SL20 genome segments. Replication of these viruses were assayed in mammalian in vitro and in vivo systems to examine replication potential in mammalian hosts. Antigenicity of inactivated vaccine candidates was then examined in chickens. Other assays such as HI and NI tests were also performed to further characterize the vaccine viruses.
Major comments
Overall: Please improve upon the organization and flow of the paper. It is important to give the reader context and provide rationale on the experimental setup. As it is written, the results are presented as unrelated sections and it is not obvious to me as to why an experiment was performed. Also, there are key information in the methods section that are needed to understand the results. Going from one section to another can be distracting for the reader. Lastly, there are many abbreviations and names that are not defined clearly, and some names are complicated and hard for a reader to remember.
Line 230-231: Before describing the results, please briefly describe the vaccination protocol used to obtain the sera used for HI assays.
- We added the brief information of vaccination protocol in lines 237 and 238 as recommended.
Line 246-271, Table 4: Have you done any statistical tests to determine differences in HI titers? Also, L226Q tends to increase avidity to RBCs, how would that affect HI titer results?
- Yes, we have done statistical analysis but there was no significant difference between groups. Although we tried to explain the effect of L226Q mutation in lines 379-402 we could not conclude how the L226Q mutation affect the lower HI titers to homologous and heterologous serum samples due to lack of direct evidence.
Line 329-330: I think that this statement is inaccurate. Since the MPMs are associated with pathogenicity in mammals, it is unclear as to whether it contributes to changes in tropism in chickens.
- We completely agree to reviewer’s comment. We replaced ‘MPMs’ with ‘mutations’ as in line 339.
Line 526-533: Can you provide an explanation as to how the dose for the inactivated viruses was determined? It appears that the viruses were not assayed (e.g. by HA) before it was administered as a vaccine. This means that the chickens received different doses of inactivated virus and thus, can be the reason for the differences observed in HI titers. If this is the case, the differences in HI titers would not be due to inherent differences in antigenicity between the vaccine candidates. In other studies, it is typical to first titer inactivated virus and normalizing HA units before using them as vaccine. HAUs are thus the units used for the vaccine dose.
- We measured virus titers (HA unit and EID50) before vaccine preparation (line 537) and treated same concentration of formaldehyde and incubation condition which is commonly used in animal vaccine industry. We did not adjust HAU after inactivation.
Minor comments
Line 73-77: While these are known MPMs, viruses isolated from patients infected with avian influenza often retain avian-like markers, suggesting that these MPMs are not absolutely necessary for a virus to infect mammals. Additionally, these MPMs often need a particular genetic background to “work” as an MPM. In other words, genetic context matters for mutations to confer phenotypic change.
- We completely agree to reviewer’s comment. To date the definition of MPMs were obscure and neglected whether they are acquired in chickens or mammals. Therefore, as the first step to discriminate we described MPMs as in lines 72-79. MPMs acquired in chickens need to be defined differently (minimal essential mutations).
Figure 2A. Please indicate error bars for the average weights
- Error bars was indicated in the Figure 2A.
Lines 101-102: I think it would be more useful to define what “poor results” refers to. E.g., does it refer to reduced virus shedding and/or clinical signs in vaccinated birds and so on.
- We revised as in lines 104 and 105.
Line 114: Please define MVV. Additionally, also avoid too many abbreviations to make the text more readable.
- We defined the MVV in line 88. Forgive us the unfamiliar abbreviations (MPMs, MVV etc.) were inevitably used due to their high frequencies of usages. We removed ‘MEMs’.
Line 123: Please provide a brief history of the SL20wt isolate. Where and which host species was it isolated from? Any clinical information regarding the bird from which SL20wt was isolated from?
- We added the brief history of the SL20wt isolate in lines 424 and 425.
Line 134-136: Why was SL20 sequence compared to that of N20-99 and SD1877? Is it because these are prototypical H9N2 viruses that are currently circulating in China and Korea?
- Since both virus had the highest homology with the genomes of SL20wt, we selected them to compare. N20-99 can be prototypic isolate in Korean but SD1844 may be one of various H9N2 viruses in China.
Line 131: Is the stalk region here of the neuraminidase protein?
- Yes, it is.
Line 279: Please provide a rationale in using r310-NS28 virus in this experiment.
- We explained the special characteristics of r310-NS28 in lines 253-256.
Line 599: Please fill out the section on Conflict of interest.
- We filled out the section on Conflict of interest.
Misc: What is the subtype of r310-NS28? If it is different, it can help explain the NI results.
- It is a Y439/Korean lineage recombinant H9N2 virus and ‘H9N2’ is added in line 427.

Reviewer 2 Report
This work is devoted to engineering by reverse genetics a vaccine against influenza caused by H9N2 viruses, that are actual for Korea. The authors have investigated various combinations of genes derived from different viral strains and H9N2 lineages and finally have found the optimal gene compatibility for vaccine strain. They have determined the importance of balance activity not only between surface protein HA and NA but also with activity of polymerase PB2. Moreover, it was shown that insertion of three mutations into PB2 have affected vaccine properties. Perfect design and well done experiments have been resulted in new interesting and important data for generation of recombinant virus strains by revers genetics.
However, there are some remarks concerning the text, that are listed below.
Sometimes there are confused words ‘gene’ and ‘genome’ that have different meaning.
LINE 340 – ‘genomes’ instead of ‘genes’
LINES: 25, 67, 100, 150, 334, 421
“internal genomes of the PR8 strain” - needs correction
It should be “internal protein genes of the PR8 strain” or “internal genomic segments of the PR8 strain”
LINES 115 - It needs insertion ‘in HA’: ‘…introduction of the L226Q mutation in HA changed…’
LINE 126-128. The data shown in these lines don’t coincide with that shown in Table S1.
‘…The nucleotide and amino acid identities to N20-99 and SD1877 were 99.29-100% and…’
Besides, there are different strain names A/chicken/Shandong/1877/2020 (SD1877) and A/chicken/Shandong/1844/2019 in text (line 125) and Table S1, respectively. Does it right?
LINES 131 – insert NA: ‘...3 amino acid deletion (63-65) in the stalk region of NA,
LINE 134 – insert ‘of NA’: ‘…in the 370 and 400 loops of 2SBS of NA tend…’
LINE 141 - Is it right ‘N-glucon’ or maybe ‘N-glucan’?
LINE 438 - BLATN replace with BLAST
Table 4, 5 - add ‘negative control’ in bottom line and ‘Dose’ in the head of the second column
Table S2. - It would be better to highlight or bold the common MPMs found in most genotype S H9N2 viruses, which are described in the text of the article (lines 129-133).
FIGURE 1. Check the coincidence between the designations *, **, ***, # and ## in Figure and in the caption.
FIGURE 3. What does it mean ‘RDE-‘ in Figure? It should be explained in a caption. For example, ‘…treated with heat (56℃, 30min; designated as RDE-) or RDE + heat (designated as RDE+) for inactivation …’ (Line 226-227).
2.6. Monovalent inactivated…
This subsection of the article needs some details of an experiment, such as name of immunized animals (mice or chicken?) and source of serum or antibodies for HI.
The results concerning PB2 are so unusual that I would like to find out some details.
What kind of amino acid does PB2 of 01310 and SL20 have in position 627? The 627E and 627K determine a host rang and pathogenicity of virus. Besides, PB2 connects with different host factors which virus uses for itself reproduction. Chicken embryo allantois (avian), cell lines MDCK (canine) and A549 (human), used for virus propagation, represent different species. Therefore, mutations both in HA and polymerase proteins can influence on difference of virus replication in these biological systems, that was observed in this work.
Could you explain whether the vaccines described in the article were developed for avian or mammals? What is your mind, does it possible to generate universal recombinant vaccine for avian and mammals?
Thanks for interesting research.
Author Response
Response to reviewer’s comments
Thank you for the precious comments and our manuscript improved apparently.
Reviewer 2
This work is devoted to engineering by reverse genetics a vaccine against influenza caused by H9N2 viruses, that are actual for Korea. The authors have investigated various combinations of genes derived from different viral strains and H9N2 lineages and finally have found the optimal gene compatibility for vaccine strain. They have determined the importance of balance activity not only between surface protein HA and NA but also with activity of polymerase PB2. Moreover, it was shown that insertion of three mutations into PB2 have affected vaccine properties. Perfect design and well done experiments have been resulted in new interesting and important data for generation of recombinant virus strains by revers genetics.
However, there are some remarks concerning the text, that are listed below.
Sometimes there are confused words ‘gene’ and ‘genome’ that have different meaning.
LINE 340 – ‘genomes’ instead of ‘genes’
LINES: 25, 67, 100, 150, 334, 421
“internal genomes of the PR8 strain” - needs correction
It should be “internal protein genes of the PR8 strain” or “internal genomic segments of the PR8 strain”
- We revised ‘internal genomes’ into ‘internal genomic segments’ in lines 23, 68, 102, 153 and 431; into ‘internal proteins’ in line 343. ‘proteins which are expressed from the seven genomes…’ in line 348 and 349.
LINES 115 - It needs insertion ‘in HA’: ‘…introduction of the L226Q mutation in HA changed…’
- We added the “in HA” in line 118.
LINE 126-128. The data shown in these lines don’t coincide with that shown in Table S1.
‘…The nucleotide and amino acid identities to N20-99 and SD1877 were 99.29-100% and…’
- We modified the data in lines 129 and 130.
Besides, there are different strain names A/chicken/Shandong/1877/2020 (SD1877) and A/chicken/Shandong/1844/2019 in text (line 125) and Table S1, respectively. Does it right?
- We revised them in lines 127 and 131.
LINES 131 – insert NA: ‘...3 amino acid deletion (63-65) in the stalk region of NA,
- We inserted “of NA” in line 134.
LINE 134 – insert ‘of NA’: ‘…in the 370 and 400 loops of 2SBS of NA tend…’
- We inserted “of NA” in line 136.
LINE 141 - Is it right ‘N-glucon’ or maybe ‘N-glucan’?
- We modified the “N-glucon” into “N-glycosylation site” in line 143.
LINE 438 - BLATN replace with BLAST
- We replaced the “BLATN” into “BLAST” in line 448.
Table 4, 5 - add ‘negative control’ in bottom line and ‘Dose’ in the head of the second column
- We added ‘negative control’ and ‘dose’ in table 4 and 5.
Table S2. - It would be better to highlight or bold the common MPMs found in most genotype S H9N2 viruses, which are described in the text of the article (lines 129-133).
- We revised Table S2 as recommended.
FIGURE 1. Check the coincidence between the designations *, **, ***, # and ## in Figure and in the caption.
- We modified the designations in Figure 1 to coincident between figure and caption.
FIGURE 3. What does it mean ‘RDE-‘ in Figure? It should be explained in a caption. For example, ‘…treated with heat (56℃, 30min; designated as RDE-) or RDE + heat (designated as RDE+) for inactivation …’ (Line 226-227).
- We added the explanation in a caption “RDE- was only heat-treated serum (56℃, 30min) or RDE + was the RDE and heat-treated serum for inactivation of non-specific inhibitors.” In Figure 3.
2.6. Monovalent inactivated…
This subsection of the article needs some details of an experiment, such as name of immunized animals (mice or chicken?) and source of serum or antibodies for HI.
- The additional information was added in 2.6 (lines 237 and 238), also in the materials and methods 4.7.
The results concerning PB2 are so unusual that I would like to find out some details.
What kind of amino acid does PB2 of 01310 and SL20 have in position 627?
- They all have glutamic acid (E) at 627.
The 627E and 627K determine a host rang and pathogenicity of virus. Besides, PB2 connects with different host factors which virus uses for itself reproduction. Chicken embryo allantois (avian), cell lines MDCK (canine) and A549 (human), used for virus propagation, represent different species. Therefore, mutations both in HA and polymerase proteins can influence on difference of virus replication in these biological systems, that was observed in this work.
- Yes, the different amino acid residues of tested recombinant viruses may influence differently in different biological systems.
Could you explain whether the vaccines described in the article were developed for avian or mammals?
- It is for animals. Although the possibility of human pandemic of avian influenza is extremely low our method may be useful to prepare vaccine strain for humans when it occurs.
What is your mind, does it possible to generate universal recombinant vaccine for avian and mammals?
- Developing real universal vaccine for animals may be difficult because they should not be inoculated with live vaccine or repeated inactivated vaccine to induce antibodies to universal epitopes in the HA2 subunit. Human live vaccines (FluMist) can induce both humoral (mucosal) and cellular immunities but they cannot be used for the elderly due to side effect. In addition, their T cell epitopes in NP and M for cross-protective cellular immunity are mismatched to contemporary seasonal flu viruses. If the residual virulence of FluMist backbones is reduced and the T cell epitopes are matched I hope more cross-protective vaccine to be developed in the near future.
Thanks for interesting research.
* We finished English editing service of MDPI.
